

# Costs of sea dikes – regressions and uncertainty estimates

Stephan Lenk[1], Diego Rybski[1], Oliver Heidrich[2], Richard J. Dawson[2], and Jürgen P. Kropp[1,3]

[1]Potsdam Institute for Climate Impact Research – PIK, Member of Leibniz Association, P.O. Box 601203, 14412 Potsdam, Germany
[2]School of Civil Engineering & Geosciences & Tyndall Centre for Climate Change Research, Newcastle University, Newcastle upon Tyne NE1 7RU, UK
[3]University of Potsdam, Institute of Earth and Environmental Science, Potsdam, Germany

*Correspondence to:* Diego Rybski (ca-dr@rybski.de)

**Abstract.** Failure to consider the costs of adaptation strategies can be seen by decision makers as a barrier for implementing coastal protection measures. In order to validate adaptation strategies to sea-level rise in the form of coastal protection, a consistent and repeatable assessment of the costs is necessary. This paper significantly extends current knowledge on cost estimates by developing, and implementing using real coastal dike data, probabilistic functions of dike costs. With the aim

of providing a reproducible estimate of typical sea dike costs and their uncertainty we analyse data from Canada and the Netherlands and relate this to published studies from the US, UK, and Vietnam. We plot the costs divided by dike length as a function of height and test four different regression models. Our analysis shows that a linear function without intercept is sufficient to model the costs, i.e. fixed costs and higher order contributions such as from the volume of core fill material are less significant. We also characterise the spread around the regression models which represents an uncertainty stemming

from factors beyond dike length and height. Drawing the analogy to project cost overruns, we employ log-normal distributions and calculate that the range between $3x$ and $x/3$ contains 95% of the data, where $x$ represents the corresponding regression value. We compare our estimates with previously published unit costs for other countries. We note that the unit costs not only depend on the country and land-use (urban/non-urban) of the sites where the dikes are being constructed but also characteristics included in the costs, e.g. property acquisition, utility relocation, project management. We provide recommendations how to

improve the reporting and estimating of the costs in order to support future adaptation studies worldwide.

## 1 Introduction

Sea-level rise represents the least uncertain consequence of climate change and there is considerable interest in comparing coastal flood damage with adaptation costs (Hallegatte et al., 2013; Boettle et al., 2013b). In line with rising temperature and sea-levels, more frequent and severe storm surges need to be anticipated (Menéndez and Woodworth, 2010). The construction

of dikes and similar defensive measures has a long history and coastal protection represents a conventional means to adapt to the threat of sea-level rise (Jonkman et al., 2013). In 1990 it was estimated that protecting 360 000 km of the coasts globally against a 1 m sea-level rise can cost at least US$ 500 billion over a 100-year period (Dronkers et al., 1990).



In order to estimate costs and benefits of adaptation to sea-level rise (Fankhauser, 1995; Jonkman et al., 2004; Dawson et al., 2009, 2011; Klijn et al., 2012; Boettle et al., 2013a; Eijgenraam et al., 2014; Jongmann et al., 2014; Kind, 2014; Kreibich et al., 2014; Boettle et al., 2016), various categories are required – in particular (i) the change of frequency and magnitude of coastal floods (Eijgenraam et al., 2014; Dawson and Hall, 2006), (ii) the expected (direct, monetary) damage without adaptation (Bouwer, 2013; Prahl et al., 2015), (iii) the residual (direct, monetary) damage with adaptation (Kreibich et al., 2014), and (iv)

the construction costs of the adaptation measure (Jonkman et al., 2013; Aerts et al., 2013).

While climate impact and adaptation research community has made progress in assessing the categories (i)-(iii), the construction costs of protection measures [category (iv)] are poorly reported in the literature. These are considered an engineering problem yet are also of utmost importance for decision makers in order to assess the scale of the investment that is required to provide protection, and is achievable for the resources available. Failure to consider the costs of strategies can invalidate,

or at least expose as impractical, recommendations from research studies. However, engineers hesitate to provide general and transferable costs. Academic literature is lacking 'real life' auditable cost information on adaptation measures (Heidrich et al., 2013) which can be of use to international, national, and local decision makers. Indeed, providing an order of magnitude and reduce uncertainties on the protection costs can remove potential barriers in designing and implementing adaptation strategies (Reckien et al., 2015; Heidrich et al., 2016).

One way to investigate construction costs is to study similar constructions of dikes that were planned or even erected in the past. Using historic construction costs are referred to as the Elemental Costs Projection approach (BCIS, 2012) but civil engineering works are profoundly influenced by various factors such as scale, nature, and characteristics of the project. This makes a straightforward use of this approach not always sensible as rates, prices, and discounts can fluctuate dramatically over short periods of time and between projects across the same regions of a country (MacDonald, 2013). For various reasons

a synthesis of coastal protection costs is very limited. On the one hand, the costs of coastal protection projects in different countries and regions are varied by socio-economic conditions like land value, land use, building prices, GDP, and general income. On the other hand, the comparability of projects is often hindered by heterogeneity and a lack of information about the site and project specifics, e.g. site preparation, site access, material sourcing etc. In addition, defence measures differ in engineering, design, and specific features like detailed dimensions and unit costs are not reported in the academic literature.

Thus, it is difficult to make generalisable estimates of the costs of a sea dike project of length $l$ and height $h$ at a given site and country.

Most authors studying adaptation to sea-level rise and adjunctive costs refer to Hoozemans et al. (1993) or Jonkman et al. (2013). The former estimated unit costs for three coastal protection measures according to the Dutch standard including dike design, construction, taxes, fees, levies, and royalties. The latter investigated costs for coastal protection of low-lying delta

areas using project-oriented case studies for the Netherlands, New Orleans, and Vietnam. The authors estimated unit costs for constructing and raising different types of hard and soft coastal protection measures. They also analysed the relationship between dike height, dike cross-section, and costs of raising dikes at a very coarse level and depending on the site. Based on data from three countries it has been suggested that the estimates from Hoozemans et al. (1993) significantly over-estimate the costs of constructing dikes (Linham et al., 2010).



Many of the above mentioned studies use *unit costs*, i.e. the dike costs can be expressed per meter height. This, however, implies the assumption that there is a linear relation without fixed costs – an assumption that to our knowledge has not been supported quantitatively so far. In addition, uncertainty is at most quantified by means of a range, i.e. some upper or lower values. This paper significantly extends these approaches by developing, and implementing using real coastal dike data, probabilistic functions of dike costs.

5    With the aim of a *reproducible* assessment of typical protection costs and their uncertainties, the work in hand explores estimated construction costs for sea dikes in Canada (Metro Vancouver) and in the Netherlands. We address two questions. First, what is the appropriate functional form for the costs of a sea dike as a function of its height? Since both, the footprint and the volume, are proportional to the height of a dike, the costs of a unit of fixed length should increase linearly and quadratically with the height, respectively, leading to the question of the composite *functional form*.

10    Second, what is the range of uncertainty that needs to be considered? Although our findings are in relation to sea dikes, our research approach and determination of uncertainty is potentially applicable and transferable to other adaptation measures. In the research on climate change adaptation, the above listed constraints of quantifying coastal protection costs represent an *uncertainty* which needs to be taken into account when cost effectiveness of adaptation measures is studied.

In order to quantify the uncertainty of the dike construction costs, we draw the parallel to project cost overruns. Construction projects usually exhibit a difference between forecasted and actual construction costs (Flyvbjerg, 2007a, b; Flyvbjerg and Stewart, 2012). Such cost overruns can have various origins, e.g. unexpected site conditions, unforeseen events, and overall underlying complexity associated with the design and construction process (Love et al., 2013). Since cost overruns represent uncertainties, which the estimator is aware of but without knowing their magnitude, they are sometimes referred to as *known-unknowns*. Chou et al. (2009) demonstrated that log-normal distributions best fit the probabilistic costs of highway bridge replacement projects. Thus, here our starting point is to characterise the deviations (spreading) around typical dike construction costs by means of a log-normal distribution.

## 2    Data

We base our main analysis on two data sources, namely *Cost of Adaptation – Sea Dikes & Alternative Strategies* (Delcan Corporation, 2012) for Canada (Metro Vancouver) and *Kosten van maatregelen – Informatie ten behoeve van het project Waterveiligheid 21e eeuw* (de Grave and Baase, 2011) for the Netherlands. The advantage is that the contained data is homogeneous (at least within each of both reports) since the cost figures come from the same sources, the same case studies, and relate to similar constructions, i.e. sea and estuarine dikes. Moreover, local conditions, e.g. affecting the exact shape to the dikes, have been taken into account.

### 2.1    Canada

Estimated costs to protect Vancouver and neighbouring municipalities against sea-level rise by 2100 are provided by Delcan Corporation (2012) – for a summary see Sec. A1. The protection measures are subdivided into 36 shoreline reaches of more





than 250 km long between the Burrard Inlet and Boundary Bay. A reach is a general term for a length of a stream or river, usually suggesting a level, uninterrupted stretch The report provides high-level long-term estimates in the preparation phase and we do not have any information if any of the planned dike has actually been constructed to date. Since it is not specified, we assume all costs are given in Canadian Dollars (CAD) 2012.

5     The costs presented in the Canadian study are referred to as a "class D estimate" in (Delcan Corporation, 2012). According to Public Works and Government Services Canada this means an indicative estimation giving unit costs and being based on a comprehensive assumptions and project requirements list. This kind of estimate is developed during the project feasibility and design stage (Public Works and Government Services Canada) and therefore subject to change.

    The Tables 4.3A and 4.3B in (Delcan Corporation, 2012) list the length and estimated costs separated for each dike so that 10  we calculate the costs per unit length. In addition, Table 2.1 in (Delcan Corporation, 2012) provides information about the expected flood levels in 2100 and the required increase of the dike height or the height of the dike to be built. Accordingly, we can study the costs per length as a function of the height, i.e. CAD per length in meters vs. height in meters. Eight of the reaches with bespoke features (e.g. barrier, see Sec. A1) have been excluded from the analysis. Additional information is included on whether the dikes are to be raised or newly constructed and whether the corresponding site is urban or rural.

15     As detailed in Sec. A1, the costs consist of (i) structural flood protection / embankment, (ii) utility relocation, pump stations, and flood boxes, (iii) property acquisition, (iv) seismic resilience measures, (v) environmental compensation, and (vi) site investigation, project management, and Engineering.

## 2.2   Netherlands

The report (de Grave and Baase, 2011) provides the estimated costs needed to raise the height of dikes across the Netherlands 20  to such an extent that the estimated risk of a flooding is decreased by ten times in comparison to the present protection level. The estimates are broken down into 205 dikes of more than 2 600 km length and the associated costs are estimated for several steps of raising the dike height (25, 50, 100, 200 cm) depending on the need (Tab. 4).

    The Tables G and K in (de Grave and Baase, 2011) list the length, the height steps, and the associated cost estimates for two different scenarios. The first scenario assumes the flood risks according to the current installed protection and the current Dutch 25  legislation. The second scenario takes the improvement of flood protection into account that is planned to be made between 2015 and 2020. According to Eijgenraam and Zwaneveld (2011) the second scenario is the more realistic one and the first one is an underestimate. As, in addition, the second scenario is based on more recent developments, we choose to use only the cost estimates of the second scenario in this study.

    As detailed in Sec. A2 the costs consist of (i) ground work and construction measures per unit of length, (ii) special measures, 30  (iii) adjustment or relocation of infrastructure, (iv) land acquisition, (v) environmental compensation, and (vi) additional costs for operations and maintenance.





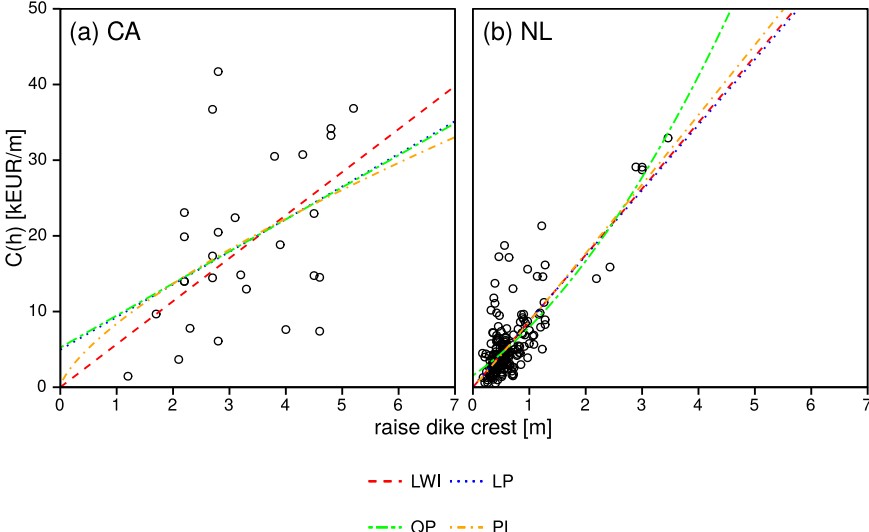

**Figure 1.** Plot of dike costs versus crest height for all dikes with a linear fit without intercept, LWI [Eq. (1), red dashed line], a linear polynomial fit, LP [Eq. (2), blue dotted line], a quadratic polynomial fit, QP [Eq. (3), green double dashed line], and a power-law fit, PL [Eq. (4), orange dotted dashed line]. (a) 28 Canadian dikes with the obtained fits and (b) 205 dikes in the Netherlands with the obtained fits. The linear and quadratic polynomials collapse in (a) and the linear without intercept with the linear polynomial in (b), see main text.

## 3 Analysis

In the following we fit regressions to the data, quantify the uncertainty, and compare our results with previous published estimates.

### 3.1 Regressions

In Fig. 1 we plot the costs per meter length of dike as a function of the raising height of the dikes. For the Canadian data [Fig. 1(a)] it can be seen, that the costs are spread over a wide range roughly between 5,000 EUR/m and 40,000 EUR/m for dike constructions of 1 m to 5 m height or raise. Only a weak tendency of reduced costs for lower heights can be guessed visually from Fig. 1. The correlation coefficient is $\varrho_{\mathrm{p}} = 0.43 \, [0.07, 0.69]$ ($[\cdot]$ denotes 95 % confidence). For the Netherlands [Fig. 1(b)] the spreading is overall smaller but most of the constructions are lower than 1 m and only few values are available for a raise of roughly 3 m. Here, the correlation coefficient is $\varrho_{\mathrm{p}} = 0.79 \, [0.74, 0.84]$.



In order to find a suitable model for typical costs we test four regressions, namely (i) linear without intercept (LWI), (ii) linear polynomial (LP), (iii) quadratic polynomial (QP), and (iv) power-law (PL):

$$\text{LWI:} \quad C(h) = bh \tag{1}$$

$$\text{LP:} \quad C(h) = a + bh \tag{2}$$

$$\text{QP:} \quad C(h) = a + bh + ch^2 \tag{3}$$

$$\text{PL:} \quad C(h) = bh^\beta \ , \tag{4}$$

where $C(h)$ (in EUR/m) are the dike costs per meter length and for height $h$, and $a, b, c, \beta$ are parameters, where $a$ denotes the
intercept (fixed costs, which are independent of the dike height), $b$ the slope, also known as *unit costs* (Hoozemans et al., 1993; Jonkman et al., 2013), $c$ the quadratic term, and $\beta$ the exponent of the power-law. The parameters have corresponding units, such as EUR/m$^3$ for $c$. For simplicity, we omit the units of the parameters. Following Hudson et al. (2015) cost elements can be well split into (i) Planning and design costs, e.g. consulting and survey costs; 2. Capital costs, e.g. enabling and construction costs; 3. Inspection costs, e.g. operational, public safety and monitoring program costs; and 4. maintenance costs, e.g. maintenance
and replacement costs.

Equation (1) is the simplest form, i.e. a linear relation with slope $b$ starting at $0\,\text{EUR/m}^2$ for $h = 0\,\text{m}$. In principle, even for very small projects, costs can emerge independent from the actual height (fixed costs), due to e.g. planning etc. Thus, Eq. (2) is similar to Eq. (1) but has an off-set, i.e. the intercept $a$, which is linked to the preparation costs. Since, the volume of the dike is proportional to the square of its height [see item 1(b) in Sec. A1], an additional quadratic term [see e.g. (Diaz, 2016)]
in Eq. (3) leads to a second-order polynomial. Finally, the power-law Eq. (4) is inspired e.g. by Eq. (4) in (Hinkel et al., 2014) and by Eq. (7) in (Fankhauser, 1995). The fitting of the regressions was carried out using non-linear least squares optimisation of the fit parameters applying the Levenberg-Marquardt algorithm (Marquardt, 1963). Using constraints of the parameters, we force the regressions to $C(0) \geq 0$ and to be monotonous increasing.

In Fig. 1(a) it can be seen for the Canadian data, that the 4 models all have similar shapes and their deviations are small
compared to the spreading of the data. It stands out that the linear and the quadratic fits collapse, i.e. the best fit of Eq. (3) is the one with a vanishing quadratic component ($c \approx 0$) so that it becomes a linear regression which is then identical to fitting Eq. (2). Since our fitting models vary strongly in the number of parameters (1...3), regressions with more parameters are expected to perform better in terms of root mean squared error (RMSE). Thus, in order to compensate for this advantage, we explore the Akaike information criterion (AIC) (Akaike, 1974) which evaluates the trade-off between the goodness of fit and
the complexity of the considered models.

In Tab. 1 the obtained fit parameters and resulting AIC and RMSE values are listed. For the Canadian data, the linear model without fixed costs, Eq. (2), performs best in terms of AIC. It is followed by the power-law model, Eq. (4), comprising the second lowest AIC value. The obtained exponent, however, indicates a curvature opposite to the one expected from a quadratic contribution due to the volume ($\beta \approx 0.7 < 1$). The linear model, Eq. (2), leads to an off-set, $a$, whose standard error is bigger
than the parameter itself, so it might be insignificant. As discussed above, the quadratic model, Eq. (3), is identical to the linear model (the quadratic contribution vanishes). It can be concluded, that the fixed costs and contributions due to the volume are





**Table 1.** Fit parameters according to Eq. (1)–(4), standard errors ($\pm$), and AIC and RMSE values for the regression types, and parameter $\sigma$ of the log-normal distributions [$\mathcal{LN}$, Eq. (5)] fitted to the residuals, for the data from Canada (Delcan Corporation, 2012) and the Netherlands (de Grave and Baase, 2011). The root mean squared error is calculated according to $\mathrm{RMSE} = \sqrt{\frac{1}{n} \sum_{i=1}^{n} \left( Y_i - \hat{Y}_i \right)^2}$, where $\hat{Y}_i$ are the fitted values and $Y_i$ are the measured ones. For both, AIC & RMSE, smaller values are better.

| fit | fit parameters | | | | AIC | RMSE | $\mathcal{LN}$ |
|---|---|---|---|---|---|---|---|
| | $a$ | $b$ | $c$ | $\beta$ | | | $\sigma$ |
| Vancouver – all dikes (# 28) | | | | | | | |
| linear, no intercept (LWI) | — | 5.7$\pm$ 0.6 | — | — | 211.5 | 9.84 | 0.62$\pm$0.08 |
| linear (LP) | 5.0$\pm$ 6.1 | 4.3$\pm$ 1.8 | — | — | 212.8 | 9.72 | — |
| quadratic (QP) | 5.2$\pm$19.1 | 4.3$\pm$12.3 | 0.0$\pm$1.8 | — | 214.8 | 9.72 | — |
| power-law (PL) | — | 8.4$\pm$ 3.4 | — | 0.7$\pm$0.32 | 212.6 | 9.69 | — |
| Vancouver – new dikes (# 14) | | | | | | | |
| linear, no intercept | — | 1.3$\pm$ 0.5 | — | — | 95.3 | 2.40 | 0.90$\pm$0.19 |
| Vancouver – raise dikes (# 14) | | | | | | | |
| linear, no intercept | — | 2.0$\pm$ 0.7 | — | — | 122.9 | 4.01 | 1.03$\pm$0.19 |
| Vancouver – urban dikes (# 21) | | | | | | | |
| linear, no intercept | — | 1.8$\pm$ 0.5 | — | — | 166.8 | 4.53 | 0.97$\pm$0.15 |
| Vancouver – rural dikes (# 7) | | | | | | | |
| linear, no intercept | — | 0.6$\pm$ 0.1 | — | — | 21.0 | 2.11 | 0.50$\pm$0.13 |
| Netherlands (# 205) | | | | | | | |
| linear, no intercept | — | 8.7$\pm$ 0.3 | — | — | 1034.6 | 3.03 | 0.54$\pm$0.03 |
| linear | 0.1$\pm$ 0.4 | 8.6$\pm$ 0.5 | — | — | 1043.5 | 3.02 | — |
| quadratic | 1.6$\pm$ 0.7 | 5.2$\pm$ 1.4 | 1.2$\pm$0.4 | — | 1038.7 | 2.98 | — |
| power-law | — | 8.7$\pm$ 0.3 | — | 1.0$\pm$0.04 | 1043.0 | 3.02 | — |

weaker than the spreading. Linear regression seems also reasonable because all quantities contribute approximately linearly to the costs, except earth fill, which contributes quadratically. The core material, however, represents a comparably small fraction of the total costs. Structural flood protection costs represent approximately 10 % of the total costs and the earth fill costs represent approximately 85 % of the former (see Sec. A1).

The regressions for the Netherlands data are shown in Fig. 1(b). Again, the fits are very similar and only the quadratic model, Eq. (3), deviates in the upper range $h > 3\,\mathrm{m}$. In Tab. 1 it can be seen, that according to AIC, the linear model without




5 off-set, Eq. (1), performs best and the quadratic model, Eq. (3), does second best. Here, the power-law, Eq. (4), and the linear model without intercept, Eq. (1), collapse. The power-law fits best for $\beta \approx 1$ so that the two models are identical, which is also reflected in the similar resulting parameters $b$. A quadratic contribution might only take effect for heights beyond the available $h$-range, which is in agreement with (Jonkman et al., 2013). Moreover, the linear model, Eq. (2), leads to a rather small off-set $a \approx 0.1$ (in particular compared to the standard error of $0.4$), so that it is almost identical to the regression without off-set.

Again, we can conclude that the linear contribution dominates and fixed costs as well as non-linear contributions from the dike volume of the can be disregarded, i.e. non-linearities are not necessary.

## 3.2 Uncertainty

While the regressions characterise the typical relation between dike height and costs, next we want to study the spreading around the fits. These deviations of the individual dikes are due to site specific properties and design features which go beyond

the height and length and are usually unavailable. Therefore, drawing the analogy with cost overruns, we employ log-normal distributions $\mathcal{LN}(\mu, \sigma)$ (Chou et al., 2009) to characterise the spreading.

Accordingly, we analyse the residuals of the fits as an estimate for the uncertainty. The residuals were calculated as ratio of the fitted values to the actually observed ones. Then we fit log-normal probability distributions

$$\mathcal{LN}(r; \mu, \sigma) = \frac{1}{\sqrt{2\pi}\sigma r} \exp\left(\frac{(\ln(r) - \mu)^2}{2\sigma^2}\right), \tag{5}$$

where $r = \frac{\hat{C}(h)}{C(h)}$ denotes the residuals, $\mu$ the location parameter, and $\sigma$ the scale parameter, using maximum likelihood estimation, and employ the Kolmogorov-Smirnov test to evaluate the goodness of the fit.

Two features support the choice of the log-normal distribution in this context. First, the log-normal distribution by definition excludes negative values. Second, the statistical spreading is relative (for the same reason). Instead of a fixed uncertainty e.g. in EUR, it is defined as a fraction, or percentage, which is plausible since bigger projects typically have larger absolute uncertainty.

For the Canadian and the Netherlands data, the estimated uncertainties are displayed together with the LWI regressions in Fig. 2. It can be seen, that the uncertainty encloses a rather large range which is increasing (due to the log-normal definition) from low to high dikes, achieving approximately 10,000 EUR–100,000 EUR for dikes of approximately 4 m–6 m height (see Tab. 1 for the obtained parameters and Sec. 3.3).

In the log-log scale (insets of Fig. 2) one can see qualitatively that the regressions and uncertainties reasonably represent

the data points. The cost estimate for raising a dike in Canada by one metre encompass roughly 6,000 EUR where the range between $3.4x$ and $x/3.4$ contains 95 % of the uncertainty ($x \approx 6,000$ EUR). Analogously, for the Netherlands, the estimate is about 9,000 EUR with an uncertainty factor 2.9. Nevertheless, few values are also located outside the 95 % ranges suggesting, that the log-normal distribution might only be a first approximation.

The Canadian data also includes information whether the dikes are completely new or if existing dikes are to be raised.

Moreover, the land-use in terms of urban/rural is specified, which strongly affects the land price (see Sec. A1) and eventually the design of the construction. Thus, we analyse the data separately according to these four categories (new/raise, urban/rural).



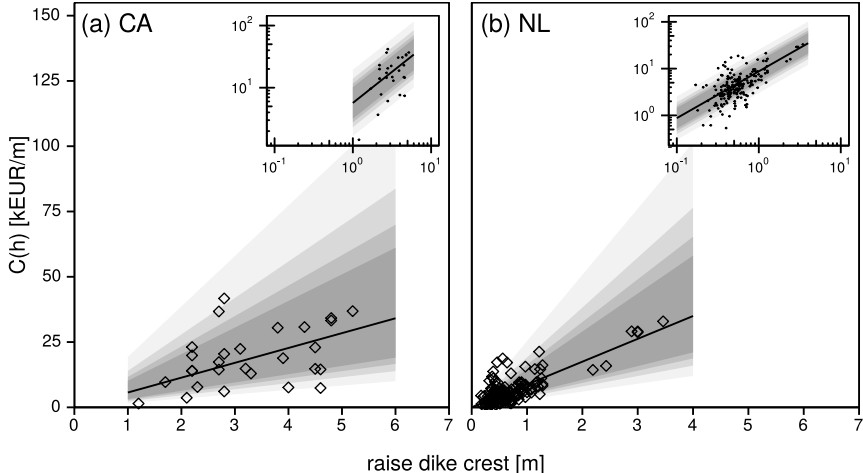

**Figure 2.** Spreading of dike costs and estimated uncertainty. The dike costs are shown together with the linear regressions without intercept, Eq. (1), and with quantiles of estimated log-normal distributions. (a) Canadian dikes and (b) dikes in the Netherlands. The insets show the same items but in double logarithmic scale. The shades give the uncertainty inclosing estimated 95%, 85%, 75%, and 65% of the values (from light to dark).

Due to the small sample size, we cannot fully disentangle all combinations, so that e.g. "new" includes both, urban and rural. We find that the fits for new and raised dikes are very similar but larger samples would be required to support this finding. In contrast, urban and rural dikes differ clearly in their costs and accordingly lead to different slopes. The resulting fit parameters, i.e. unit costs and log standard deviations, are also included in Tab. 1. For the Netherlands a similar difference between urban and rural has been reported by de Grave and Baase (2011). Mostly the land value is causing the discrepancy between urban and rural dikes. The actual land value however is very site specific and is highly sensitive to land-use and socio-economic changes as discussed earlier.

### 3.3 Comparison with results from other studies

To compare our results with results from Hoozemans et al. (1993) and Jonkman et al. (2013) we converted all data to EUR for the Netherlands in 2012, involving three adjustments, namely currency, purchasing power, and reference year.

The easiest way of adjusting different currencies would be to use the exchange rates. However, this does not take into account differences in purchasing power. Considering these, the exchange rates have to be adjusted. To do so, a method, used e.g. by the World Bank, is to adjust the exchange rate of a currency $i$ to US dollars (USD) $e_{i,\mathrm{USD}}$ by correcting it using purchasing power parities (PPP). The World Bank provides both the gross domestic product (GDP) and the purchasing power parity adjusted GPD (GDP$_{\mathrm{PPP}}$) for the countries of the world (World Bank, 2016). Using these, a correction factor $c_e$ for the exchange rate can be derived by calculating the ratio of the GDP and the GDP$_{\mathrm{PPP}}$ of a country according to $c_e = \mathrm{GDP}_{\mathrm{PPP}}/\mathrm{GDP}$, leading to an evaluation of how much currency $i$ is over- or under-rated in comparison to USD. Then the given exchange rate $e_{i,\mathrm{USD}}$



**Table 2.** GDP per capita, purchasing power parity (PPP) GDP per capita, and mean exchange rates ($e_{i,\mathrm{USD}}$) of 2012 (World Bank, 2016) for Canada, the Netherlands, and the USA, as well as the PPP adjusted exchange rates $e^*_{i,\mathrm{USD}}$ and $e^*_{i,\mathrm{EUR}}$. The PPP adjusted exchange rates give the amount of money in the currency of the country having the same purchasing power as 1.00 USD and 1.00 EUR, respectively.

|    | GDP [USD] | GDP$_{\mathrm{PPP}}$ [PPP-\$] | $e_{i,\mathrm{USD}}$ [USD$^{-1}$] | $e^*_{i,\mathrm{USD}}$ [PPP-\$$^{-1}$] | $e^*_{i,\mathrm{EUR}}$ [EUR$^{-1}$] |
|----|-----------|-------------|------------|-------------|-------------|
| CA | 52 733.5  | 42 280.8    | 1.00 CAD   | 1.25 CAD    | 1.51 CAD    |
| NL | 49 128.1  | 46 053.9    | 0.78 EUR   | 0.83 EUR    | 1.00 EUR    |
| US | 51 456.7  | 51 456.7    | 1.00 USD   | 1.00 USD    | 1.20 USD    |
| GB | 41 294.5  | 37 607.9    | 0.63 GBP   | 0.69 GBP    | 0.83 GBP    |

can be corrected according to $e^*_{i,\mathrm{USD}} = e_{i,\mathrm{USD}}/c_e$ by dividing it by the correcting factor $c_e$. This gives the amount of money of currency $i$ one has to spend to have the same purchasing power as one USD. For converting the PPP adjusted exchange rate $e^*_{i,\mathrm{USD}}$ into EUR, it is divided by the PPP adjusted exchange rate $e^*_{\mathrm{EUR,USD}}$ from EUR into USD, which can be derived as described just before. This leads to the result of $e^*_{\mathrm{CAD,EUR}} = 1.51$ and $e^*_{\mathrm{USD,EUR}} = 1.20$ for 2012, meaning that CAD 1.51 and USD 1.20 have the same purchasing power as 1.00 EUR in the Netherlands in 2012. Using these factors, the cost estimates given in CAD and USD can be converted into EUR.

For the data from Hoozemans et al. (1993) the USD prices of 1993 were adjusted to prices of 2012 first, by using implicit price deflator values for the GDP of the USA from the US Federal Reserve Bank (Federal Reserve Bank of St. Louis, 2016). The price deflator index is set to $D_{2009} = 100$ for 2009. For 1993 it is $D_{1993} = 72.244$ and for 2012 it is $D_{2012} = 105.231$. The cost for 2012 is calculated according to $C_{2012} = (C_{1993} \cdot D_{2012})/D_{1993}$. Then these values were converted into EUR by dividing them by the PPP adjusted exchange rate $e^*_{\mathrm{USD,EUR}}$ of Tab. 2.

Jonkman et al. (2013) give the costs in EUR of 2009. To ensure comparability, these cost estimates where adjusted to values of 2012 by applying the same procedure using price deflators given by the US Federal Reserve Bank of St. Louis (Federal Reserve Bank of St. Louis, 2016). So the costs for 2012 $C_{2012}$ is given by $C_{2012} = (C_{2009} \cdot D_{2012})/D_{2009}$, where $C_{2009}$ is the cost given by Jonkman et al. (2013) and $D_{2009} = 99.2$ and $D_{2012} = 101.6$.

We also include results for Great Britain by using the dimensions of the dikes designed in the Canadian study to parameterise the calculation tool developed for the Environment Agency (Pettit and Robinson, 2012). We entered the dike dimensions and, utilising the 80th percentile of the costs, obtained estimations of the costs, if the dikes specified in (Delcan Corporation, 2012) (Sec. 2.1) would be built in Great Britain.

In Fig. 3 we compare the various values reported in the literature with our estimates in terms of $b$ the slope known as unit costs as described earlier. According to the available information, we separate dikes constructed on urban and rural land, and those where land use is not specified. Moreover, we colour-code the 5 countries for which we have information, i.e. the Netherlands, Canada, the USA, Vietnam, and Great Britain. Where available, ranges are plotted and error bars (for 95 % uncertainty). It can be seen that the cost estimates by Hoozemans et al. (1993) are not in all cases underestimates as suggested by Jonkman



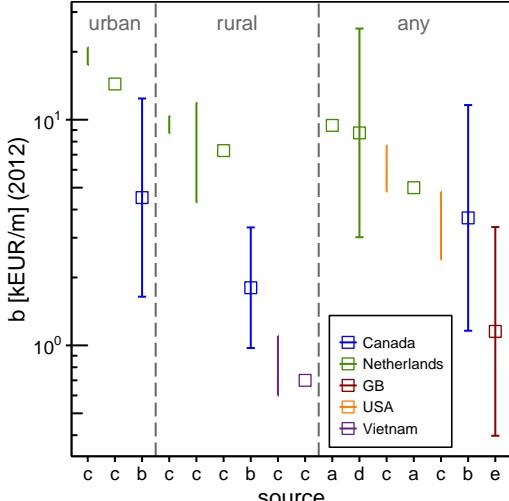

**Figure 3.** Comparison of our results with data from different studies and countries in terms of unit cost estimates for raising dikes. Namely, a – (Hoozemans et al., 1993), b – this study based on (Delcan Corporation, 2012), c – (Jonkman et al., 2013), d – this study based on (de Grave and Baase, 2011), and e – (Pettit and Robinson, 2012) using data from (Delcan Corporation, 2012). Boxes represent single unit cost estimates, plain bars given ranges, and the boxes with error bars our calculations with mean and 95% quantiles of the log-normal distributions. To make the data comparable, the values have been adjusted to Euros of 2012.

et al. (2013). They line up quite well within the range of the other cost estimates for the Netherlands of de Grave and Baase (2011) and Jonkman et al. (2013). Moreover, it can be seen that the difference between the land uses is smaller than between the countries. Nevertheless, when considering only the individual countries there are still differences between the land uses (as expected, urban dikes tend to be more expensive). But overall it can be said that the costs of dikes constructed in the Netherlands is the highest, followed by costs for dikes built in the USA, Canada, and Vietnam.

The cost estimate for Great Britain differs quite largely from that for Canada. It is substantially lower and roughly on the level of Vietnam. One reason could be that, although taking into account also a fixed percentage for investment, operation, and maintenance costs, the approach of the Canadian study is more detailed leading to a higher cost estimate. Another reason could be, that the PPP-corrected GDP of Great Britain is about 11% lower than that of Canada leading to lower costs.

## 4 Summary and Discussion

To reduce one of the potential barriers of planning and implementing adaptation strategies in relation to coastal dikes (Heidrich et al., 2016; Reckien et al., 2015) we offer a rigorous empirical basis for the assumption of constant unit costs (i.e. linear with height, no fixed costs). Moreover, we characterise the deviations of these typical costs by means of log-normal distributions assuming the relative deviations are independent of the height. This variability captures influences which go beyond the two strongest factors, namely length and height. Although the data from the Netherlands covers mostly dikes, or dike raises, of less





than 1 m, whereas the data from Canada includes dikes as high as 7 m, the uncertainty is of similar order of magnitude, i.e. in both cases the multiplication with and division by a factor of 3 includes approximately 95%.

Nevertheless, we do find neither statistical signatures of fixed costs nor of non-linearities for the costs of dikes we have analyses. For both coastal protection projects, i.e. in Canada and the Netherlands, it is sufficient to express the typical costs of dikes per height and length, which is compatible with assumptions made in previous publications, e.g. in (Jonkman et al., 2013).

We offer a rigorous empirical basis for the assumption of constant unit costs (i.e. linear with height, no fixed costs). Moreover, we characterise the deviations of these typical costs by means of log-normal distributions assuming the relative deviations are independent of the height. This variability captures influences which go beyond the two strongest factors, namely length and height. Although the data from the Netherlands covers mostly dikes, or dike raises, of less than 1 m, whereas the data from Canada includes up to almost 7 m, the uncertainty is of similar order of magnitude, i.e. in both cases the multiplication with

and division by a factor of 3 includes approximately 95 %.

The uncertainty considered here stems from a lack of knowledge, i.e. by studying the costs only as a function of length and height, as we did not have more detailed information on the local conditions and requirements of the dikes available (although we assume that the authors of the reports did have a better knowledge). Hence, we borrow the concept of cost overruns to characterise the uncertainty of dike constructions. Worth noting, that when erected, the constructions may be affected by real

cost overruns (the original Canadian study included 50 % contingency), which would increase both, the overall costs and the spreading. Thus, in particular in regard of the uncertainty, our results are probably only lower estimates. Another aspect to be mentioned is that we assume all dikes to have equal probability, i.e. each dot in Fig. 1 is equally likely. However, which dike design and corresponding costs are required depends on the local topography and their likelihood is characteristic to the case study. Moreover, we are not accounting for any economic shock that would affect the costs, e.g. raw material and fuel costs but

also labour shocks, shocks affecting imports, etc.

We also want to discuss another aspect that comes into play, when the total costs of an ensemble of dikes are aggregated, e.g. according to $C_\text{tot} = \sum_i l_i C(h_i)$, where $i$ are the indices of considered dikes. Due to the central limit theorem, the standard deviation decreases with the square root of the sample size. If the costs of the dikes are independent from each other, then one can expect that the relative uncertainty will decrease with the number of dikes. In reality, however, it can be expected that the

costs of the various dikes are correlated so that the relative uncertainty of the total costs is likely not to shrink [Prahl et al. (2012) treat spatial correlations in a different context].

Comparing our estimates with the figures provided in the literature, we find that the costs differ more between the countries than between the land uses. Nevertheless, within the countries the differences from the land uses are still pronounced and as expected, urban dikes tend to be more costly than rural ones. However, comparing such cost estimates it is crucial which

components are actually included in the figures. While in the Canadian report (Delcan Corporation, 2012) all components of the costs are disclosed in other cases it might not be clear if e.g. costs of property acquisition or project management are included, or if they refer to the pure costs of the physical construction.

To conclude, this study gives decision makers an order of magnitude on the protection costs which can remove potential barriers in designing and implementing adaptation strategies worldwide. Future research may focus on the creation of a "best





practice" approach to understand how potential impacts are accounted for and to deliver decision makers ways in which climate

adaptation options such as sea dikes are understood and measured, both in terms of investment needed economically but also

in reducing risks of flooding and reduced damage costs.

## Appendix A: Explicit cost estimates

### A1 Canada

The total costs [from Table 4.2 in (Delcan Corporation, 2012)] consist of

1. Structural flood protection / embankment

   9 % of total costs

   For a dike of length $l$, footprint $f$, height $h$ the following components are included, whereas '$\sim$' represents proportionality.

   (a) Site preparation: clearing and removal of topsoil (costs/$l \sim ht$, where $t$ is the thickness of the layer), estimated unit

cost CAD 15 /m$^3$.

   (b) Core material: supplying and installing the dike material (costs/$l \sim fh$, approximately costs/$l \sim h^2$), estimated

   unit cost CAD 40 /m$^3$.

   (c) Rip-rap: stone protection for the water side of the dike (costs/$l \sim ht$, where $t$ is the thickness of the layer), estimated

   unit cost CAD 50 /m$^3$.

(d) Surface restoration: construction of a typical asphalt road in case there is already a road at the site, applies to 5/36

   reaches (costs/$l \sim f \sim h$, assuming surface $\sim lf$), estimated unit cost CAD 100 /m$^2$.

2. Utility relocation, pump stations, and flood boxes

   4 % of total costs

   (a) It was assumed that dike construction will include 25 % extra in urban areas and 5 % for rural areas for utility

relocation.

   (b) Upgrades of existing pump stations. This applies to 16/36 reaches, with an estimated unit cost of CAD 2.5 million.

   (c) Adjustment of drainage behind the dike and small pump station installation. This applies to 18/36 reaches, with an

   estimated unit cost of CAD 0.5 million.

3. Property acquisition

17 % of total costs

   The area of property to be acquired is determined by the footprint, i.e. costs/$l \sim f$, approximately costs/$l \sim h$. Full

   purchase costs have been included, i.e. as derived from previous constructions. Residential property has double the value

   of commercial/industrial property and commercial/industrial property has five times the value of agricultural property.

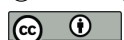



(a) agricultural: 3 % of total costs, 9/36 reaches, 86 % of total area, estimated unit cost CAD 22 /m$^2$;

(b) residential: 6 % of total costs, 7/36 reaches, 4 % of total area, estimated unit cost CAD 850 /m$^2$;

(c) commercial/industrial: 8 % of total costs, 11/36 reaches, 11 % of total area, estimated unit cost CAD 400 /m$^2$.

4. Seismic resilience (vibro-replacement, deep soil mixing, toe berm)

   34 % of total costs

   Because the Vancouver area is seismic active, it is necessary to make dikes seismically resilient. Depending on the soil profile at the dike location vibro-replacement, deep soil mixing or installing a toe berm is necessary.

   (a) vibro-replacement: 10/36 reaches, estimated unit cost CAD 22 /m$^2$;

   (b) deep soil mixing: 3/36 reaches, estimated unit cost CAD 250 /m$^2$;

   (c) toe berm: 10/36 reaches.

5. Environmental compensation

   1 % of total costs, 4/36 reaches, estimated unit cost CAD 250 /m$^2$.

6. Site Investigation, Project Management, and Engineering (15 % on top of previous items)

   2 % of total costs

7. Contingency (50 % on top of all previous items)

   33 % of total costs

Prior to beginning the analysis, we perform a few steps:

– exclude the following reaches:
  #4 (floodwall), #5 (flood proofing, no information), #10 (barrier), #16 (double dike), #17 (flood proofing, no information), #23 (retreat), #27 (barrier), #28 (flood proofing, no information);

– disregard deep soil mixing of reaches #7, #8, #22; and

– disregard 50% contingency on top of the total costs.

This leads to a total of 28 reaches being analysed. The complete data is provided in Tab. 3.

**A2   Netherlands**

The dike raising costs of the Dutch study are based on a system of cost functions. To obtain the cost function for one dike reach there are eight calculation steps taken into account:

1. Identify the needed dike height raising by modelling the hydraulical strain for a given dike height and return level interval.



**Table 3.** Cost estimates after (Delcan Corporation, 2012) for Metro Vancouver without contingency for 28 dikes (out of 36). Total costs have been estimated using the reach length and increased dike height and the unit costs given in (Delcan Corporation, 2012).

| reach number | reach name | action | land use | increase [m] | reach length [m] | unit costs [kEUR/m] |
|---|---|---|---|---|---|---|
| 1A | South Vancouver | new | urban | 2.2 | 3 245 | 11.7 |
| 1B | South Vancouver | new | urban | 2.2 | 11 325 | 11.7 |
| 2 | Burnaby | new | urban | 2.8 | 7 710 | 17.1 |
| 3 | Queensborough | raise | urban | 3.1 | 7 190 | 18.7 |
| 6 | Richmond Urban / high density | raise | urban | 2.7 | 9 015 | 14.5 |
| 7 | Richmond Rural / low density / north | raise | rural | 2.8 | 11 440 | 5.1 |
| 8 | Richmond Rural / low density / south | raise | rural | 2.3 | 16 190 | 6.5 |
| 9 | Richmond West dike | raise | urban | 4.5 | 6 390 | 19.2 |
| 11A | Sea Island | raise | urban | 3.9 | 4 850 | 15.7 |
| 11B | Sea Island | raise | urban | 2.2 | 10 550 | 16.6 |
| 12 | Tilbury/Sunbury | raise | urban | 2.2 | 15 450 | 19.3 |
| 13 | Ladner | raise | urban | 2.7 | 4 300 | 30.7 |
| 14A | Westham Island | new | rural | 4.6 | 4 560 | 12.1 |
| 14B | Westham Island | new | rural | 3.3 | 6 940 | 10.8 |
| 15 | Delta West dike | raise | rural | 4.5 | 8 840 | 12.3 |
| 18 | Boundary Bay Village | new | urban | 4.8 | 1 215 | 28.6 |
| 19 | Boundary Bay Regional Park | new | rural | 4.6 | 2 205 | 6.2 |
| 20 | Beach Grove | raise | urban | 4.8 | 1 165 | 27.8 |
| 21 | Boundary Bay | raise | rural | 4.0 | 14 775 | 6.4 |
| 22 | Surrey | raise | urban | 2.8 | 7 150 | 14.1 |
| 24 | Crescent Beach | raise | urban | 3.8 | 2 590 | 25.5 |
| 25 | Annacis Island | new | urban | 2.7 | 13 550 | 12.1 |
| 26 | Kitsilano and English Bay | new | urban | 2.1 | 1 280 | 3.0 |
| 29 | West Vancouver | new | urban | 5.2 | 7 300 | 30.8 |
| 30 | District of North Vancouver | new | urban | 1.7 | 5 800 | 8.1 |
| 31 | City of North Vancouver | new | urban | 3.2 | 2 000 | 12.4 |
| 32 | Port Moody | new | urban | 1.2 | 875 | 1.3 |
| 33 | White Rock / South Surrey | new | urban | 4.3 | 2 500 | 25.7 |

2. Process the information about the current and the required dike profile to determine the needed ground and construction measures. This step takes into account:

   (a) the dike height of the initial situation,





(b) the benching height of the initial situation if there is benching on the land side,

(c) the distance between the outer dike crest and the dike foot on the land side,

(d) the required raising of the base body of the dike (corresponding to the required raising of the dike crest including an additional raising for settling and compaction),

(e) broadening of the dike base for increasing macrostability,

(f) broadening of the dike base for piping.

3. Determination of the range of ground and construction measures according to a four step expulsion list with the following combinations of measures:

M1  complete solution with exclusively ground measures,

M2  raising and fortification of the dike body with a combination of ground measures and one construction measure on the dike toes on the land side,

M3  dike raising in the ground and steepening of the (inside) embankment on one side in combination with one construction measure within the dike body,

M4  dike raising in the ground and steepening of both dike embankments in combination with a cofferdam construction within the dike body.

According to the selected combination of measures, the new dike profile, the type and extent of the required construction measures, the additional footprint of the dike, the direct ground work and construction costs per unit of length, the length of the dike section to which the measures are applied, the total construction costs and the additional costs for administration and maintenance are calculated.

4. Calculation of costs needed for dike reaches with special conditions. These special conditions consist of the construction of a cofferdam. The costs for this is estimated using both the horizontal and vertical length, the height and a standard cost function.

5. Estimation of costs needed for the adjustment of infrastructure. This applies if there is an existing road or other type of traffic infrastructure which has to be reallocated. This also applies if there are crossroads or railways located above the dike which interfere with the raising of the dike. This may require the construction of a new dike section.

6. Identification of costs for purchase of land. To estimate this kind of costs there are four cases which are being differentiated, namely built-up area, non built-up area, urban and rural. For the two categories built-up and urban the land acquisition costs are considered to be high and for non built-up and rural comparatively low.

7. Determination of costs for countryside and environment compensation measures. If a dike reach crosses a nature reserve or an area of special scenic importance, it is necessary to acquire land create appropriate compensation measures.





8. Estimation of the volume of the total investment costs and additional annual costs for administration and maintenance. The total investment costs are formed by summing up the costs (including their administration and maintenance costs) of the previous seven steps. Based on this, the total administration and maintenance costs are defined as percentage of the total investment costs.

The complete data is provided in Tab. 4.

*Acknowledgements.* We thank H. Costa, and B. Prahl for useful discussions. The research leading to these results has received funding from the European Community's Seventh Framework Programme under Grant Agreement No. 308497 (Project RAMSES).





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





**Table 4.** Cost estimates after de Grave and Baase (2011) for the Netherlands adjusted for 2012. The cost estimates given here are according to the second reference situation in (de Grave and Baase, 2011). The unit costs represent the costs needed to build a dike of the length of one meter with the given height.

| reach code | reach name | length [m] | increase [m] | unit costs [kEUR/m] | reach code | reach name | length [m] | increase [m] | unit costs [kEUR/m] |
|---|---|---|---|---|---|---|---|---|---|
| 1-1-1 | Schiermonnikoog | 3 930 | 0.65 | 4.7 | 29-1-2 | Walcheren-West | 750 | 0.86 | 9.6 |
| 2-1-1 | Ameland | 16 590 | 0.64 | 4.3 | 29-2-1 | Walcheren-Oost | 20 840 | 0.90 | 8.9 |
| 3-1-1 | Terschelling | 14 000 | 0.54 | 3.6 | 30-1-1 | Zuid-Beveland-West | 24 190 | 1.06 | 6.8 |
| 4-1-1 | Vlieland | 1 340 | 0.52 | 5.4 | 30-1-2 | Zuid-Beveland-West | 15 970 | 0.87 | 7.2 |
| 5-1-1 | Texel | 1 000 | 0.65 | 3.1 | 30-1-3 | Zuid-Beveland-West | 14 290 | 0.44 | 2.9 |
| 5-1-2 | Texel | 25 100 | 0.65 | 4.7 | 30-1-4 | Zuid-Beveland-West | 8 000 | 0.30 | 1.7 |
| 6-1-1 | Friesland-Groningen-Lauwersmeer | 2 700 | 0.78 | 4.2 | 31-1-1 | Zuid-Beveland-Oost | 20 130 | 1.28 | 8.9 |
| 6-1-2 | Friesland-Groningen-Lauwersmeer | 8 950 | 0.80 | 7.7 | 31-1-2 | Zuid-Beveland-Oost | 21 200 | 0.60 | 2.9 |
| 6-2-1 | Friesland-Groningen-Groningen | 57 390 | 0.88 | 7.1 | 31-1-3 | Zuid-Beveland-Oost | 7 290 | 0.30 | 3.5 |
| 6-2-2 | Friesland-Groningen-Groningen | 27 440 | 0.92 | 6.9 | 32-1-1 | Zeeuwsch Vlaanderen-West | 21 590 | 1.08 | 6.9 |
| 6-3-1 | Friesland-Groningen-NoordFriesland | 72 470 | 0.74 | 5.0 | 32-1-2 | Zeeuwsch Vlaanderen-West | 3 010 | 1.14 | 14.6 |
| 6-4-1 | Friesland-Groningen-IJselmeer | 14 790 | 0.58 | 3.6 | 32-1-3 | Zeeuwsch Vlaanderen-West | 1 600 | 1.17 | 9.6 |
| 6-4-2 | Friesland-Groningen-IJselmeer | 9 450 | 0.32 | 1.4 | 32-1-4 | Zeeuwsch Vlaanderen-West | 3 440 | 1.19 | 9.8 |
| 6-4-3 | Friesland-Groningen-IJselmeer | 9 650 | 0.32 | 2.9 | 32-1-5 | Zeeuwsch Vlaanderen-West | 2 650 | 1.26 | 11.2 |
| 6-4-4 | Friesland-Groningen-IJselmeer | 19 060 | 0.40 | 0.5 | 32-1-6 | Zeeuwsch Vlaanderen-West | 530 | 1.22 | 21.3 |
| 6-4-5 | Friesland-Groningen-IJselmeer | 8 700 | 0.55 | 4.8 | 32-1-7 | Zeeuwsch Vlaanderen-West | 760 | 0.91 | 5.4 |
| 7-1-1 | Noordoostpolder | 31 540 | 0.97 | 3.9 | 32-2-1 | Zeeuwsch Vlaanderen-Oost | 19 860 | 1.28 | 16.2 |
| 7-1-2 | Noordoostpolder | 13 000 | 0.85 | 3.0 | 32-2-2 | Zeeuwsch Vlaanderen-Oost | 24 760 | 1.28 | 8.3 |
| 7-1-3 | Noordoostpolder | 11 240 | 0.25 | 1.3 | 34-1-1 | West-Brabant | 26 720 | 0.61 | 4.4 |
| 8-1-1 | Flevoland-Noordoost | 5 150 | 0.72 | 5.6 | 34-1-2 | West-Brabant | 10 670 | 0.63 | 4.6 |
| 8-1-2 | Flevoland-Noordoost | 12 420 | 1.00 | 5.8 | 34-1-3 | West-Brabant | 9 900 | 0.42 | 3.9 |
| 8-1-3 | Flevoland-Noordoost | 17 560 | 1.08 | 5.7 | 34a-1-1 | Geertrudenberg | 2 760 | 0.73 | 2.6 |
| 8-1-4 | Flevoland-Noordoost | 5 190 | 0.93 | 8.5 | 34a-1-2 | Geertrudenberg | 7 180 | 0.42 | 3.6 |
| 8-2-1 | Flevoland-ZuidWest | 23 380 | 0.57 | 5.0 | 35-1-1 | Donge | 16 260 | 0.61 | 2.4 |
| 8-2-2 | Flevoland-ZuidWest | 5 620 | 0.43 | 3.1 | 35-1-2 | Donge | 12 270 | 0.42 | 6.8 |
| 8-2-3 | Flevoland-ZuidWest | 25 540 | 0.45 | 3.0 | 36-1-1 | Land van Heusen de Maaskant | 63 240 | 0.51 | 1.8 |
| 9-1-1 | Vollenhove | 8 010 | 0.38 | 2.3 | 36-1-2 | Land van Heusen de Maaskant | 17 230 | 0.55 | 3.7 |
| 9-1-2 | Vollenhove | 34 730 | 0.40 | 2.0 | 36-1-3 | Land van Heusen de Maaskant | 3 900 | 0.56 | 4.5 |
| 10-1-1 | Mastenbroek | 3 730 | 0.56 | 2.5 | 36-1-4 | Land van Heusen de Maaskant | 11 370 | 0.53 | 4.5 |
| 10-1-2 | Mastenbroek | 11 160 | 0.43 | 4.2 | 36-1-5 | Land van Heusen de Maaskant | 1 000 | 0.51 | 2.0 |
| 10-1-3 | Mastenbroek | 19 480 | 0.35 | 3.5 | 36-1-6 | Land van Heusen de Maaskant | 2 780 | 0.50 | 7.4 |
| 10-1-4 | Mastenbroek | 13 350 | 0.39 | 3.5 | 36-1-7 | Land van Heusen de Maaskant | 5 000 | 0.51 | 3.1 |
| 11-1-1 | IJseldelta | 4 690 | 1.23 | 5.0 | 36a-1-1 | Keent | 4 400 | 0.51 | 1.4 |
| 11-1-2 | IJseldelta | 21 380 | 0.42 | 6.2 | 37-1-1 | Nederhermert | 1 360 | 0.54 | 3.0 |
| 11-1-3 | IJseldelta | 6 810 | 0.37 | 4.4 | 38-1-1 | Bommelerwaard-Waal | 29 580 | 0.56 | 6.0 |
| 12-1-1 | Wieringen | 11 610 | 0.84 | 4.5 | 38-2-1 | Bommelerwaard-Maas | 2 520 | 0.54 | 2.8 |
| 12-1-2 | Wieringen | 20 690 | 0.41 | 1.8 | 38-2-2 | Bommelerwaard-Maas | 10 510 | 0.57 | 3.7 |
| 13-1-1 | Noord-Holland-Noord | 12 430 | 0.72 | 6.9 | 38-2-3 | Bommelerwaard-Maas | 2 160 | 0.58 | 6.6 |
| 13-1-2 | Noord-Holland-Noord | 2 630 | 0.64 | 17.1 | 38-2-4 | Bommelerwaard-Maas | 5 020 | 0.56 | 5.5 |
| 13-1-3 | Noord-Holland-Noord | 5 610 | 0.67 | 5.8 | 39-1-1 | Alem | 4 750 | 0.51 | 5.8 |
| 13-1-4 | Noord-Holland-Noord | 5 680 | 2.43 | 15.9 | 40-1-1 | Heerenwaarden-Waal | 5 280 | 0.45 | 2.5 |
| 13-2-1 | Noord-Holland-Westfriesland | 26 430 | 0.41 | 5.2 | 40-2-1 | Heerenwaarden-Maas | 6 380 | 0.63 | 3.9 |
| 13-2-2 | Noord-Holland-Westfriesland | 29 920 | 0.41 | 4.7 | 41-1-1 | Land van Maas en Waal-Waal | 40 670 | 0.58 | 6.5 |
| 13-4-1 | Noord-Holland-Waterland | 24 730 | 0.36 | 7.2 | 41-1-2 | Land van Maas en Waal-Waal | 990 | 0.58 | 4.1 |
| 13-4-2 | Noord-Holland-Waterland | 15 300 | 0.32 | 5.8 | 41-2-1 | Land van Maas en Waal-Maas | 43 810 | 0.52 | 3.5 |
| 13a-1-1 | Noord-Holland-Waterland | 1 530 | 0.21 | 4.0 | 41-2-2 | Land van Maas en Waal-Maas | 480 | 0.54 | 3.7 |
| 13a-1-2 | Noord-Holland-Waterland | 6 420 | 0.26 | 1.0 | 42-1-1 | Ooij en Millingen | 8 080 | 0.59 | 4.7 |
| 13a-1-3 | Noord-Holland-Waterland | 3 260 | 0.26 | 0.6 | 42-1-2 | Ooij en Millingen | 9 350 | 0.59 | 6.0 |
| 13a-1-4 | Noord-Holland-Waterland | 1 630 | 0.21 | 0.6 | 42-1-3 | Ooij en Millingen | 42 600 | 0.59 | 4.7 |
| 13b-1-1 | Marken | 8 630 | 0.29 | 2.6 | 43-1-1 | Betuwe, Tieler- en C'waarden | 16 470 | 0.39 | 3.5 |
| 14-1-1 | Zuid-Holland-Kust | 2 820 | 3.00 | 29.1 | 43-1-2 | Betuwe, Tieler- en C'waarden | 7 680 | 0.38 | 2.8 |
| 14-1-2 | Zuid-Holland-Kust | 1 640 | 3.00 | 28.7 | 43-1-3 | Betuwe, Tieler- en C'waarden | 18 320 | 0.35 | 2.7 |
| 14-1-3 | Zuid-Holland-Kust | 1 400 | 3.46 | 32.9 | 43-1-4 | Betuwe, Tieler- en C'waarden | 17 020 | 0.34 | 3.0 |
| 14-1-4 | Zuid-Holland-Kust | 5 000 | 2.89 | 29.1 | 43-1-5 | Betuwe, Tieler- en C'waarden | 4 500 | 0.30 | 1.8 |
| 14-2-1 | Zuid-Holland-NweWaterweg-West | 4 590 | 0.97 | 8.7 | 43-1-6 | Betuwe, Tieler- en C'waarden | 10 790 | 0.41 | 4.4 |
| 14-3-1 | Zuid-Holland-NweWaterweg-Oost | 26 030 | 0.84 | 8.6 | 43-1-7 | Betuwe, Tieler- en C'waarden | 7 450 | 0.56 | 3.3 |
| 14-3-2 | Zuid-Holland-NweWaterweg-Oost | 10 150 | 0.71 | 13.0 | 43-1-8 | Betuwe, Tieler- en C'waarden | 41 640 | 0.60 | 7.8 |
| 15-1-1 | Lopiker- en Krimpenerwaard | 23 060 | 0.35 | 4.1 | 43-1-9 | Betuwe, Tieler- en C'waarden | 46 600 | 0.58 | 6.2 |
| 15-1-2 | Lopiker- en Krimpenerwaard | 2 500 | 0.39 | 11.1 | 44-1-1 | Kromme Rijn-Rijn | 32 480 | 0.36 | 2.6 |
| 15-1-3 | Lopiker- en Krimpenerwaard | 17 210 | 0.32 | 10.2 | 44-2-1 | Kromme Rijn-Meren | 15 630 | 0.35 | 3.1 |
| 15-1-4 | Lopiker- en Krimpenerwaard | 4 850 | 0.39 | 13.7 | 44-2-2 | Kromme Rijn-Meren | 3 250 | 0.23 | 4.4 |
| 16-1-1 | Alblasserwaard en de Vijfheerenlanden | 32 230 | 0.43 | 6.4 | 44-2-3 | Kromme Rijn-Meren | 4 700 | 0.55 | 2.4 |
| 16-1-2 | Alblasserwaard en de Vijfheerenlanden | 16 660 | 0.45 | 6.2 | 45-1-1 | Gelderse Ballei-Rijn | 5 350 | 0.37 | 2.7 |
| 16-1-3 | Alblasserwaard en de Vijfheerenlanden | 10 590 | 0.45 | 8.0 | 45-2-1 | Gelderse Ballei-Maren | 9 920 | 0.79 | 4.3 |
| 16-1-4 | Alblasserwaard en de Vijfheerenlanden | 17 350 | 0.46 | 17.2 | 45-2-2 | Gelderse Ballei-Maren | 7 520 | 0.24 | 2.0 |
| 16-1-5 | Alblasserwaard en de Vijfheerenlanden | 9 390 | 0.42 | 9.8 | 45-2-3 | Gelderse Ballei-Maren | 10 390 | 0.18 | 1.3 |
| 17-1-1 | IJselmonde | 24 800 | 0.58 | 5.6 | 46-1-1 | Eempolder | 1 100 | 0.56 | 5.6 |
| 17-1-2 | IJselmonde | 8 560 | 0.44 | 1.0 | 46-1-2 | Eempolder | 7 310 | 0.17 | 4.5 |