# Peer review of "Costs of sea dikes – regressions and uncertainty estimates"

_Natural Hazards and Earth System Sciences, 2016_

## Referee Comment (RC1) · Anonymous Referee #1 · 14 Sep 2016

In their manuscript, Lenk et al. use data on the cost of sea-dike construction from the Netherlands and Canada with the aim of producing typical unit costs for use in further studies. Such an effort is very much required, and therefore I believe this paper to be of potentially great value to the field. There has been a great deal of attention in recent years for the needs to assess adaptation costs, yet this aspect remains very much undercovered in the scientific literature compare to studies on the damages caused by natural hazards. The information in this paper can help researchers to start to make the important step towards also including the costs of the adaptation in their analyses. I believe that the paper is well-written, timely, succinct and therefore I would recommend publication subject to the author's addressing several points below.

• The lack of good quality costs estimates has hampered progress in the advance of large scale flood risk modelling. Whilst many advances have been made in assessing

large scale flood risk (e.g. Hinkel et al., 2014; Ward et al., 2013; Arnell and Gosling, 2014), these studies either do not address the costs of adaptation, or do so in a simplified way (e.g. Hinkel et al., 2014). The need for data on the location and costs of dikes is called for also in a recent commentary by Ward et al. (2016). It would be useful to mention how the current manuscript can support such efforts (in the introduction or the discussion), which would make further explicit the wide implications of this paper.

• The abstract states that the paper provides recommendations on how to improve the reporting and estimating of the costs (of sea-dikes) in order to support future adaptation studies worldwide. However, I did not really find anything in the manuscript to warrant such a statement; I was expecting some vision of how a database should be further developed. Whist I don't think this is required in this paper, I believe that this statement in the abstract should be dropped.

• I think that the paper would be improved by a more explicit discussion of how operation and maintenance costs are included in the datasets. The paper is intended to provide some kind of information and guidance for assessing adaptation costs. In terms of sea-dikes, O&M costs can be extremely high over the lifetime of a dike, perhaps as high as the investment costs. How should decision-makers deal with this? Are there rules of thumbs that can be prescribed, like taking a percentage of investment costs per year to represent O&M costs?

• Similar discussions to those sketched here are taking place in the field of river flood modeling. It would be worth mentioning to what extent this research is / is not useful for such studies.

Small comments • Page 2, line 6. Insert "the" between "While" and "climate" (i.e. "While the climate. . .") • Page 2, line 16: Replace "Using historic construction costs are. . ." with "Using historic construction costs is. . ." • Page 3, line 27. "Replace ". . .affecting the exact shape to the dikes" with ". . .affecting the exact shape of the dikes". • Page 12, line 3: "Nevertheless, we do find neither statistical signatures of

fixed costs nor of non-linearities for the costs of dikes we have analyses." This sentence does not flow well, I am not sure what point the authors are trying to make.

References Arnell, N.W., Gosling, S.N., 2014. The impacts of climate change on river flood risk at the global scale. Climatic Change, 134, 387-401.

Hinkel et al., 2014. Coastal flood damage and adaptation costs under 21st century sea-level rise. PNAS, 111, 3292-3297.

Ward, P.J. et al., 2013. Assessing flood risk at the global scale: Model setup, results, and sensitivity. Environmental Research Letters, 8, 044019.

Ward, P.J., Jongman, B., Salamon, P., Simpson, A., Bates, P., De Groeve, T., Muis, S., Coughlan de Perez, E., Rudari, R., Trigg, M.A., Winsemius, H.C., 2015. Usefulness and limitations of global flood risk models. Nature Climate Change, 5, 712-715, doi:10.1038/nclimate2742.

---

## Referee Comment (RC2) · Anonymous Referee #2 · 18 Oct 2016

The authors apply an empirical approach to derive a functional form for the costs of construction of coastal dikes as a function of height. With this purpose, they set up several regression models with costs as predictand and several. They estimate the parameters of these models using cost data from the Netherlands and Canada. The main conclusion is that, somewhat surprisingly, the costs can be mainly described by a liner function of dike height, with no (or very small) fixed costs.

My general evaluation of the manuscript is positive, although there are some issues that the authors may want to address in a revised version.

1. One general point is that the English needs some editing. This does not deter from understanding the current version, but the text will benefit from copy-editing before eventual publication.

[Figure]

2. One main concern is that the authors quite freely extrapolate the results obtained from these two regions to offer recommendations 'worldwide'. I think this can be an oversimplification, especially the conclusion that the fixed costs seem to be so low. Will this conclusion also apply to other regions with a weaker tradition of adaptation to sea-level rise or to areas that so far have not been threatened by sea level rise so far? I can imagine that in those regions the fixed costs can be substantial.

Some particular points

3. 'Sea-level rise represents the least uncertain consequence of climate change and there is considerable interest in comparing coastal flood damage with adaptation costs

It is not clear what the authors mean with this sentence. The range of estimations of future sea-level rise is very broad and i depends on physical mechanism, like land-ice dynamics, that is not well understand and that it is actually not implemented in current climate models. Perhaps the authors mean that it is certain that global sea-level will rise, but the magnitude and its regional distribution is very uncertain, much more than temperature. There may be areas where sea-level will fall or rise, for instance in the Northern Hemisphere at high latitudes, depending on the rate of melting of Greenland and Antarctica

4. 'The latter investigated costs for coastal protection of low-lying delta areas using project-oriented case studies for the Netherlands, New Orleans, and Vietnam.'

'The latter' is here too unspecific , as there is a sentence in between. It is not clear whom the yuathors are referring to.

5. 'They also analysed the relationship between dike height, dike cross-section, and costs of raising dikes at a very coarse level and depending on the site'

analyzed at a very coarse level and depending on the site the relationship... As it stands now the sentence could mean that the dikes were built at very coarse level

6. 'Second, what is the range of uncertainty that needs to be considered? '

I think the authors mean what are the sources of uncertainty that need to be considered

7. 'The report provides high-level long-term estimates in the preparation phase and we do not have any information if any of the planned dike has actually been constructed to date'

what does 'high-level' mean here

8. ' provides information about the expected flood levels in 2100'

is this the expected relative mean sea-level rise or extreme sea-level rise events ? if the latter,. which percentile?

9. 'dike constructions of 1 m to 5 m height or raise'

of raise ? the sentence is unclear

10. 'In Fig. 1(a) it can be seen for the Canadian data, that the 4 models all have similar shapes and their deviations are small' delete comma after data

11.'Fig. 2. It can be seen, that the uncertainty encloses a rather large range which is increasing (due to the log-normal definition)'

delete comma after seen

12. 'The cost estimate for raising a dike in Canada by one metre encompass roughly 6,000 EUR '

encompass is not the right word here, I think

13. 'Nevertheless, few values are also located outside the 95 % ranges suggesting, that the log-normal distribution might only be a first approximation.'

delete comma after suggesting. The word 'also' is irritating here. Do the authors mean 'In addition, few values are...

14. 'Moreover, it can be seen that the difference between the land uses is smaller than

between the countries. '

This is a somewhat sloppy wording. The authors mean that the influence of land use is smaller than the influence of country
* * *

---

## Author Comment (AC1) · 10 Jan 2017

1. In their manuscript, Lenk et al. use data on the cost of sea-dike construction from the Netherlands and Canada with the aim of producing typical unit costs for use in further studies. Such an effort is very much required, and therefore I believe this paper to be of potentially great value to the field. There has been a great deal of attention in recent years for the needs to assess adaptation costs, yet this aspect remains very much undercovered in the scientific literature compare to studies on the damages caused by natural hazards. The information in this paper can help researchers to start to make the important step towards also including the costs of the adaptation in their analyses. I believe that the paper is well-written, timely, succinct and therefore I would recommend publication subject to the author's addressing several points below.

> We appreciate the positive evaluation of our manuscript.

2. The lack of good quality costs estimates has hampered progress in the advance of large scale flood risk modelling. Whilst many advances have been made in assessing large scale flood risk (e.g. Hinkel et al., 2014; Ward et al., 2013; Arnell and Gosling, 2014), these studies either do not address the costs of adaptation, or do so in a simplified way (e.g. Hinkel et al., 2014). The need for data on the location and costs of dikes is called for also in a recent commentary by Ward et al. (2016). It would be useful to mention how the current manuscript can support such efforts (in the introduction or the discussion), which would make further explicit the wide implications of this paper.

> We agree with the reviewer (also a similar point was raised by RC2) and would add a paragraph that the transfer of our findings and conclusions to other countries needs to be done with caution. Although plausible, we have no prove that the analogous parameters and consequent conclusions hold true in other countries. Especially it cannot be excluded that fixed costs could represent a significant contribution in countries with weaker tradition in coastal protection or in countries that so far have not been threatened by sea-level rise. Similar arguments could also apply to the unit costs. Further research will be necessary to better understand the unit costs and potential fix costs so that they can be transferred to arbitrary countries. In the context of riverine floods, the lack of good quality cost estimates has hampered progress in the advance of large scale flood risk modelling. There is an urgent need for data on the location of dikes as well as their costs and uncertainty (Ward et al. 2015). Given the information of flood management in place (i.e. existing protection levels), this need could be fulfilled by automatised identification of the required protection courses, so that our results could help to estimate dike costs. Certainly, coastal and riverine dikes have different requirements in particular regarding wave run-up and duration of floods. However, it is plausible that our main conclusions regarding linear cost function and uncertainty also apply to riverine dikes.

3. The abstract states that the paper provides recommendations on how to improve the

reporting and estimating of the costs (of sea-dikes) in order to support future adaptation studies worldwide. However, I did not really find anything in the manuscript to warrant such a statement; I was expecting some vision of how a database should be further developed. Whist I don't think this is required in this paper, I believe that this statement in the abstract should be dropped.

> We agree and would drop the statement from the abstract (similar point was raised by R2).

4. I think that the paper would be improved by a more explicit discussion of how operation and maintenance costs are included in the datasets. The paper is intended to provide some kind of information and guidance for assessing adaptation costs. In terms of sea-dikes, O&M costs can be extremely high over the lifetime of a dike, perhaps as high as the investment costs. How should decision-makers deal with this? Are there rules of thumbs that can be prescribed, like taking a percentage of investment costs per year to represent O&M costs?

> We agree and would add a paragraph to the discussion. Operation and maintenance is another example. Maintenance costs can vary significantly and over an order of magnitude (Keating et al. 2015). Such costs depend on frequency of inspections, annual maintenance requirements, and long-term intermittent maintenance activities(Keating et al. 2015). Typical activities include repairs, vegetation cutting, weed and vermin control, and others. While the Netherlands data does include maintenance, the Canadian data and the Great Britain estimates do not. The annual operation costs are usually given by a percentage of the construction costs over the lifetime of the dike. If we assume 1% of annual maintenance costs over a lifetime of 100years, then maintenance adds up to 100% of the construction costs (disregarding any discounting) so that maintenance can represent a substantial contribution to the costs. The reason why maintenance sometimes is included and sometimes not, is probably budgeting. In terms of accounting, construction costs usually represent one-time expenses and operation costs spread out over many years.

5. Similar discussions to those sketched here are taking place in the field of river flood modeling. It would be worth mentioning to what extent this research is / is not useful for such studies.

> Again, we agree and would discuss this issue together with above mentioned paragraph around (Ward et al. 2015).

6. Small comments - Page 2, line 6. Insert "the" between "While" and "climate" (i.e. "While the climate. . .") - Page 2, line 16: Replace "Using historic construction costs are. . ." with "Using historic construction costs is. . ." - Page 3, line 27. "Replace ". . .affecting the exact shape to the dikes" with ". . .affecting the exact shape of the dikes".

> We appreciate finding these mistakes. All mistakes will be corrected.

7. - Page 12, line 3: "Nevertheless, we do find neither statistical signatures of fixed costs nor of non-linearities for the costs of dikes we have analyses." This sentence does not flow well, I am not sure what point the authors are trying to make.

> We agree would simplify this sentence.

References Arnell, N.W., Gosling, S.N., 2014. The impacts of climate change on river flood risk at the global scale. Climatic Change, 134, 387-401. Hinkel et al., 2014. Coastal flood damage and adaptation costs under 21st century sea-level rise. PNAS, 111, 3292-3297. Ward, P.J. et al., 2013. Assessing flood risk at the global scale: Model setup, results, and sensitivity. Environmental Research Letters, 8, 044019. Ward, P.J., Jongman, B., Salamon, P., Simpson, A., Bates, P., De Groeve, T., Muis, S., Coughlan de Perez, E., Rudari, R., Trigg, M.A., Winsemius, H.C., 2015. Usefulness and limitations of global flood risk models. Nature Climate Change, 5, 712-715, doi:10.1038/nclimate2742.

> We thank the referee for providing these references and refer to these at appropriate places within the paper.

---

## Author Comment (AC2) · 10 Jan 2017

A. The authors apply an empirical approach to derive a functional form for the costs of construction of coastal dikes as a function of height. With this purpose, they set up several regression models with costs as predictand and several. They estimate the parameters of these models using cost data from the Netherlands and Canada. The main conclusion is that, somewhat surprisingly, the costs can be mainly described by a liner function of dike height, with no (or very small) fixed costs.

> We appreciate this insightful comment and agree with the surprising conclusion. As outlined in the manuscript, higher-order components contribute to the costs so that the linear function is indeed somewhat surprising. Other colleagues commonly use unit costs and implicitly make the assumption of a linear relationship. Our results give

empirical support for this assumption.

B. My general evaluation of the manuscript is positive, although there are some issues that the authors may want to address in a revised version.

> We appreciate the positive evaluation.

1. One general point is that the English needs some editing. This does not deter from understanding the current version, but the text will benefit from copy-editing before eventual publication.

> We are sorry that the English is not of sufficient quality. We will ensure another native speaker proof reads this.

2. One main concern is that the authors quite freely extrapolate the results obtained from these two regions to offer recommendations 'worldwide'. I think this can be an oversimplification, especially the conclusion that the fixed costs seem to be so low. Will this conclusion also apply to other regions with a weaker tradition of adaptation to sea-level rise or to areas that so far have not been threatened by sea level rise so far? I can imagine that in those regions the fixed costs can be substantial.

> We agree with the reviewer. As suggested by RC1 we would drop the last sentence of the abstract where recommendations worldwide were mentioned. Moreover, we would add a paragraph to the summary discussing the limitations of the transferability of our findings and parameters.

Some particular points

3. 'Sea-level rise represents the least uncertain consequence of climate change and there is considerable interest in comparing coastal flood damage with adaptation costs. It is not clear what the authors mean with this sentence. The range of estimations of future sea-level rise is very broad and i depends on physical mechanism, like land-ice dynamics, that is not well understand and that it is actually not implemented in current climate models. Perhaps the authors mean that it is certain that global sea-level will

rise, but the magnitude and its regional distribution is very uncertain, much more than temperature. There may be areas where sea-level will fall or rise, for instance in the Northern Hemisphere at high latitudes, depending on the rate of melting of Greenland and Antarctica.

> The reviewer makes a good point. Our statement could be misleading. Accordingly, we would change it to "Sea-level rise represents a foreseeable consequence of climate change and there is considerable interest in comparing coastal flood damage with adaptation costs".

4. 'The latter investigated costs for coastal protection of low-lying delta areas using project-oriented case studies for the Netherlands, New Orleans, and Vietnam.' 'The latter' is here too unspecific , as there is a sentence in between. It is not clear whom the yuathors are referring to.

> We agree and would simply fix the problem by providing the reference at the end of each sentence.

5. 'They also analysed the relationship between dike height, dike cross-section, and costs of raising dikes at a very coarse level and depending on the site' analyzed at a very coarse level and depending on the site the relationship... As it stands now the sentence could mean that the dikes were built at very coarse level.

> We thank the referee for finding this flaw and would change it accordingly.

6. 'Second, what is the range of uncertainty that needs to be considered? ' I think the authors mean what are the sources of uncertainty that need to be considered

> Here we disagree with the referee. We actually mean the range in the sense as mentioned eg in the abstract, ie the range enclosing 95% of spreading. But as this seem to be ambiguous that could lead to a misunderstanding we will insert a clarifying statement to that effect.

7. 'The report provides high-level long-term estimates in the preparation phase and we

do not have any information if any of the planned dike has actually been constructed to date' what does 'high-level' mean here?

> This terminology was taken from the Canadian report (p.25). But we agree with the referee that it is not clear. Accordingly, we would drop "high-level long-term".

8. ' provides information about the expected flood levels in 2100' is this the expected relative mean sea-level rise or extreme sea-level rise events ? If the latter,. which percentile?

> The report is based on the "Sea Dike Guidelines" by Ausenco Sandwell (2011) and takes into account sea-level rise, subsidence, reference tide, storm surge, wind set-up, and wave height. Different return periods are mentioned for storm surges, estuarine reaches, and seismic design. For our work it is secondary how the necessary crest heights have been designed, we explore the provided crest heights and corresponding costs. Accordingly, we prefer not to confuse the reader by providing too many details.

9. 'dike constructions of 1 m to 5 m height or raise' of raise ? the sentence is unclear

> We agree and would rephrased the sentence: "For the Canadian data ... it can be seen that the costs are spread over a wide range roughly between 5,000EUR/m and 40,000EUR/m for dike construction or raise with final heights of 1m to 5m."

10. 'In Fig. 1(a) it can be seen for the Canadian data, that the 4 models all have similar shapes and their deviations are small' delete comma after data

> We agree.

11.'Fig. 2. It can be seen, that the uncertainty encloses a rather large range which is increasing (due to the log-normal definition)' delete comma after seen

> We agree.

12. 'The cost estimate for raising a dike in Canada by one metre encompass roughly 6,000 EUR ' encompass is not the right word here, I think

> We agree and will simplify the statement by stating 'The cost for raising a dike in Canada by one meter is estimated to be 6,000 EUR.'

13. 'Nevertheless, few values are also located outside the 95 % ranges suggesting, that the log-normal distribution might only be a first approximation.' delete comma after suggesting. The word 'also' is irritating here. Do the authors mean 'In addition, few values are...

> We agree and would remove the comma as well as "also".

14. 'Moreover, it can be seen that the difference between the land uses is smaller than between the countries. ' This is a somewhat sloppy wording. The authors mean that the influence of land use is smaller than the influence of country

> We agree and would change the manuscript accordingly.